# Exploring women's development group leaders' support to maternal, neonatal and child health care: A qualitative study in Tigray region, Ethiopia

Fisseha Ashebir[1,2], Araya Abrha Medhanyie[2], Afework Mulugeta[2], Lars Åke Persson[3,4], Della Berhanu[3,4]*

1 Tigray Regional Health Bureau, Mekelle, Tigray, Ethiopia, 2 School of Public Health, College of Health Sciences, Mekelle University, Mekelle, Tigray, Ethiopia, 3 London School of Hygiene & Tropical Medicine, London, United Kingdom, 4 Ethiopian Public Health Institute, Addis Ababa, Ethiopia

* della.berhanu@lshtm.ac.uk

## Abstract

### Background

Community health workers and volunteers are vital for the achievement of Universal Health Coverage also in low-income countries. Ethiopia introduced community volunteers called women's development group leaders in 2011. These women have responsibilities in multiple sectors, including promoting health and healthcare seeking.

### Objective

We aimed to explore women's development group leaders' and health workers' perceptions on these volunteers' role in maternal, neonatal and child healthcare.

### Methods

A qualitative study was conducted with in-depth interviews and focus group discussions with women's development group leaders, health extension workers, health center staff, and woreda and regional health extension experts. We adapted a framework of community health worker performance, and explored perceptions of the women's development group program: inputs, processes and performance. Interviews were recorded, transcribed, and coded prior to translation and thematic analysis.

### Results

The women's development group leaders were committed to their health-related work. However, many were illiterate, recruited in a sub-optimal process, had weak supervision and feedback, lacked training and incentives and had weak knowledge on danger signs and care of neonates. These problems demotivated these volunteers from engaging in maternal, neonatal and child health promotion activities. Health extension workers faced difficulties in managing the numerous women's development group leaders in the catchment area.

**Data Availability Statement:** The qualitative data cannot be open access due to confidentiality reasons. De-identified data will be stored on s

secure sever at the Ethiopian Publica Health Institute and can be made available upon request by contacting Martha Zeweldemariam martha.zeweldemariam@lshtm.ac.uk.

**Funding:** This work was supported, in whole by the Bill & Melinda Gates Foundation [INV-009691]. Under the grant conditions of the Foundation, a Creative Commons Attribution 4.0 Generic License has already been assigned to the Author Accepted Manuscript version that might arise from this submission. The funder had no role in the study design, collection, management, analysis, and or interpretation of data. https://www.gatesfoundation.org/.

**Competing interests:** The authors have declared that no competing interests exist.

## Conclusion

The women's development group leaders showed a willingness to contribute to maternal and child healthcare but lacked support and incentives. The program requires some redesign, effective management, and should offer enhanced recruitment, training, supervision, and incentives. The program should also consider continued training to develop the leaders' knowledge, factor contextual influences, and be open for local variations.

## Introduction

Community health worker programs often adhere to the 1978 Alma-Ata Declaration's principles of enhancing community involvement in the planning and implementation of their primary healthcare [1, 2]. The majority of low- and middle-income countries have implemented such programs, recognizing that community health workers link the community with the health system [3–5]. In the era of the Millennium Development Goals, the community health worker programs were essential parts of the primary healthcare [6, 7]. These programs have continued to be vital towards achieving universal health coverage as part of the Sustainable Development Goals [8]. Community health workers particularly play a major role in reducing maternal [9], neonatal and child deaths [10]. Despite their significant role in providing promotive, preventative and, at times, curative health services, they frequently have poor knowledge and little training in maternal, newborn, and child health [11, 12]. Reportedly they may have insufficient supervision [4], lack incentives [13, 14] and recognition [15], and suffer from weak linkages with the health system and other stakeholders [16]. Sometimes the recruitment process is at fault, leading to lack of community trust [5, 16]. All of these issues may affect the community health workers' performance.

Ethiopia has implemented different community health worker programs over five decades [17]. In 2003, the Health Sector Development Program introduced the Health Extension Program to deliver primary healthcare through a community approach [18]. As part of this program, 34,000 Health Extension Workers were deployed to rural communities [10]. There are usually two health extension workers at each heath post serving a population of 3000–5000 people, providing maternal, neonatal, and child health services as part of their tasks [17]. The Women's Development Groups (WDG) with volunteer leaders were partly established to support and be a link to the maternal, newborn, and child health services at the health post. The WDG comprises groups of 25 up to 30 women residing in one neighborhood. One of the members is selected to lead the group (1–30 leaders). These groups are further divided into smaller groups of six women, led by one leader and is referred to as the 1 to 5 network [16]. The WDG has been given different names over time, including Women's Development Army, Women's Development Team Army, social mobilizers, and Health Development Army [16].

The Ethiopian Government took different steps to strengthen the WDG strategy [10, 17]. From 2010 to 2015, the WDG strategy mobilized women and was claimed to contribute to the increase in skilled assistance at delivery from 10% to 26% [18, 19]. The WDGs were engaged in a range of activities that in some cases could negatively influence their household income [20, 21]. We have shown that WDG leaders' health-related knowledge was low, particularly in newborn and child health [16]. These leaders reported engaging with pregnant women, but were less active in promoting newborn and child health. The reported levels of interaction with health extension workers, women in their networks, and key community stakeholders were also suboptimal.

There is a need to listen to the voices of the WDG leaders regarding their support to maternal, newborn and child health in their local context. The health extension workers at the health posts and in the communities are their immediate links to the health system. Studies of the WDG leaders' situation and opinions are lacking [16, 20]. Hence, using qualitative methods and a framework of community health workers' performance [22], this study aimed to explore WDG leaders' and health extension workers' perceptions of these volunteers' contributions to maternal, neonatal, and child healthcare in Tigray region, Ethiopia.

## Methods

### Study setting

Tigray region is located in the northern part of Ethiopia. The study was conducted in two woredas (districts) of Tigray region: Enderta woreda, located in the South East zone of Tigray, and Kilte-Awlaelo woreda, located in the East zone of Tigray. Enderta has a population of 125,739 and six primary healthcare units with 12 satellite health posts. Kilte-Awlaelo has a population of 111,993 (2017/18: CSA, projected from 2007 census) and five primary healthcare units with 18 satellite health posts [23]. Like most rural communities in Ethiopia, the populations of the study areas live on subsistence farming and have limited access to basic healthcare.

This study was part of a broader project to evaluate the Optimization of the Health Extension Program Intervention [24]. The intervention aimed to increase service utilization by under-five children. A baseline survey that was part of the evaluation was conducted from December 2016-February 2017 in 52 districts of which eight were rural woredas in Tigray. We have previously reported on WDG leaders based on that survey [16]. In order to capture a range of factors that may affect WDG leaders' performance, we used data from that survey to select one woreda where WDG leaders had good knowledge on the promotion of maternal, neonatal, and child health, and one woreda with low knowledge. Fieldwork was done from July 2018 to August, 2018.

### Study participants

We used purposive sampling to select study participants. With the assistance of health extension workers, 1–30 and 1 to 5 WDG leaders were selected from the two districts of Tigray region that were included in the previous survey. To be included in the study, the WDG leaders had to serve a large number of women, be active in their role, and be able to explain their ideas. In addition, we included health extension workers who supported the selected WDG leaders, health extension worker supervisors at health centers serving them, woreda maternal, neonatal and child health and Health Extension Program experts from study woredas, and a Health Extension Program coordinator from the regional health bureau (Table 1).

**Table 1. Description of participants, Tigray region, Ethiopia, 2018.**

| Study site | Number of in-depth interviews | | Number of focus groups | | Number of key informant interviews | | |
|---|---|---|---|---|---|---|---|
| | 1 to 30 WDG leaders | 1 to 5 WDG leaders | WDG leaders | HEWs | HEW supervisors | Woreda experts | Regional expert |
| Enderta | 4 | 4 | 1 | 1 | 1 | 2 | |
| Kilte-Awlaelo | 4 | 3 | 1 | 1 | 1 | 2 | |
| Tigray health bureau | 0 | 0 | 0 | 0 | 0 | 0 | 1 |
| Total | 8 | 7 | 2 | 2 | 2 | 4 | 1 |

WDG = women's development group; HEW = health extension worker

## Data collection

The first author (FA) and three research assistants with Masters in Public Health from the School of Public health, College of Health Science, Mekelle University, who were experienced in qualitative data collection, were trained for one day by the first author (FA). The training included a detailed review of the study protocol, topic guide and mock interviews.

The topic guides were developed in English (S1 File), translated into the local language, Tigrigna (S2 File), pre-tested and modified. Data were collected in key informant interviews, in-depth interviews, and focus group discussions (Table 1). The topic guides included exploration of the participants' experiences and opinions on WDG inputs, programmatic processes, and contribution to the primary healthcare system, especially maternal neonatal, and child health. The tools were continuously revised based on emerging themes during data collection.

Seven key informant interviews, 15 in-depth interviews, and four focus group discussions were carried out. Each interview took on average one hour. Every focus group included 7–9 participants with a mean duration of one and a half hours. The interviews were conducted in Scheduled areas to provide privacy and avoid noise. All interviews were conducted in Tigrigna and were audio recorded with the permission from study participants. Audio recordings were transcribed verbatim and, after that, translated from Tigrigna into English.

## Data analysis

During data collection, interview notes were read and re-read. This provided an iterative process to reflect and be immersed in the details and specifics of the data, allowing for the discovery of important patterns, themes, and interrelationships, which were followed up in subsequent interviews. All transcripts were imported into ATLAS.TI 7.5.4 software for coding and analysis. Inductive data analysis was guided by three broad tasks: data reduction, data display and conclusion drawing or verification [25]. The transcripts were re-read, and coded until saturation of the emerging themes was achieved. Text searches were also carried out for pertinent key words and quotations. After debriefing sessions to compare findings by the study team members, interpretations were confirmed and related to the adapted framework for measuring community health workforce performance within primary health care systems [22] with the following thematic areas: 1) Inputs (logistics and commodities); 2) Programmatic processes (supportive systems, WDG development, support from community based groups); and 3) Community health systems performance outputs (WDG competency and WDG well-being) (Fig 1) and revision of the national guidelines to the WDG program [16]. The final outcome, improved maternal, neonatal, and child health, was beyond the scope of this study.

## Ethical considerations

Ethical approval was obtained from the Institutional Review Board (IRB) of Mekelle University, College of Health Sciences (protocol number 1433/2018). Support letters were also obtained from the Tigray regional health bureau and woreda health offices. Verbal consent was obtained from all participants, and the use of verbal consent was approved by the ethical committee. Privacy and confidentiality was maintained during interviews and discussions. Participants were also given identification numbers to conceal their identities. All audio recordings and translated notes were stored on password-protected files accessed only by the researcher.

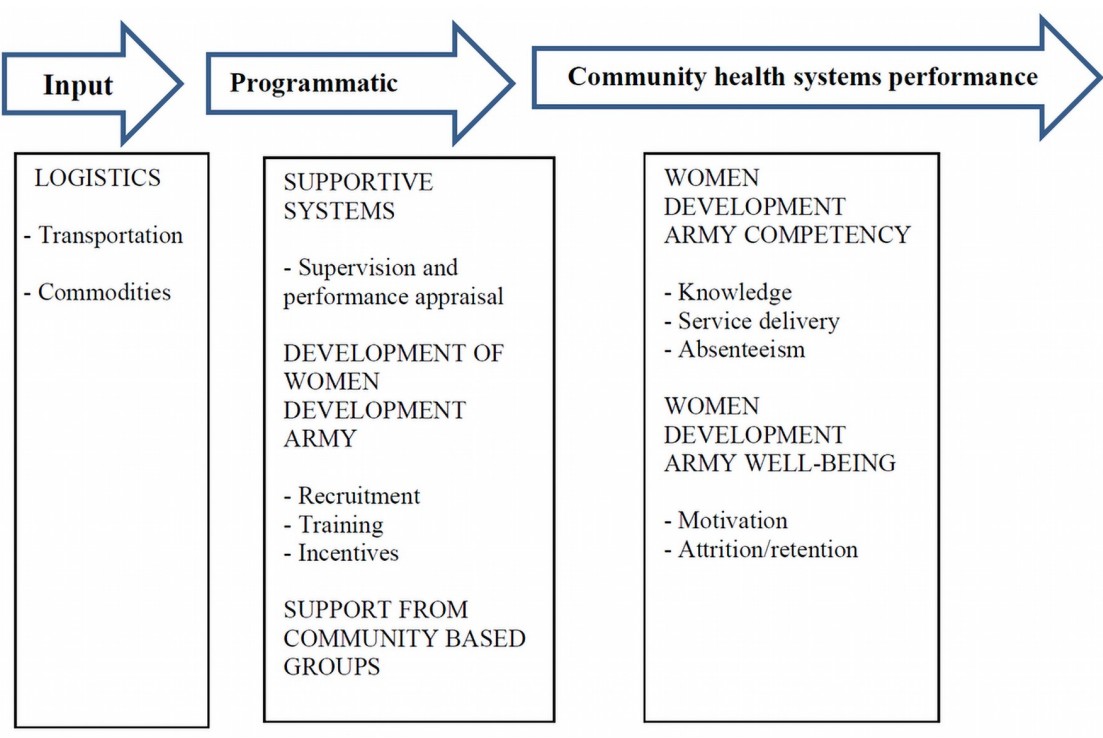

**Fig 1. Adapted framework for the community health worker performance measurement.**

## Results

### Background and socio-demographic distribution of participants

Fifteen in-depth interviews, four focus group discussions and seven key informant interviews were conducted. Thirty-one WDG leaders, sixteen health extension workers, two of their supervisors, and four woreda and one regional health experts were interviewed. Of a total of thirty-one WDG leaders, twenty-two were married, nineteen were above 35 years old, and eighteen were illiterate. Fourteen of the respondents interviewed had been working as WDG leaders five years or more (Table 1).

### Inputs

**Logistics.** *Transportation and commodities.* WDG leaders mentioned that there was insufficient logistic support to their maternal, neonatal and child health work. Long distances, absence of transportation, and poor mobile networks were frequently mentioned to negatively affect their performance. These problems created barriers to their health service support, in particular to WDG leaders who promoted deliveries at health facilities. The absence of user-friendly reading material, including job aids, hindered their health promotion activities.

"*How could I teach others? I wouldn't lie in this regard. I assume that educating others does not belong to me.*"

(In-depth interview, 1 to 30 WDG leader)

A WDG leader also reported that in the last two years nobody had provided any material to fulfill her roles.

*I am using paper and pen from my children's school materials to send written reports to the health extension worker.*"

(Focus group discussion, a 1 to 30 WDG leader)

## Programmatic processes

**Supportive systems.** *Supervision and performance appraisal.* A few WDG leaders reported that they received support from health extension workers, while a majority of the leaders frequently claimed that they were not supported and supervised.

"*We gather our members and conduct the gathering without the health extension workers.*"

(In-depth Interview, a 1 to 30 WDG leader)

All respondents reported a low commitment of staff at health posts and health centers as a key obstacle to provide supportive supervision of WDG leaders. The health extension workers mentioned their own problems with lack of job satisfaction, limited opportunities for career advancement, the fatigue they had developed after years of working and their age. These issues inhibited their support and follow-up to WDG leaders. The health extension workers also reported that they had too many WDG leaders to support, supervise and appraise.

"*The proportion of women members to a WDG leader and WDG leaders to HEW is almost over the limit now.*"

(Key informant interview, health extension worker supervisor)

Furthermore, district-level primary healthcare experts reported gaps in their own supportive supervision of WDG leaders and health extension workers.

"*When the program started there were individuals from higher levels who supervised our work, supported us on how to do more, and taught us. But now, there is no support at all.*"

(In-depth interview, a 1 to 30 WDG leader)

In addition, primary healthcare experts from district and regional levels called for a supervision manual. Such a tool could have counteracted the weak supervision and performance appraisal of the WDG leaders.

**WDG leaders' development.** *Recruitment.* The nomination process of WDG leaders was conducted by village women's associations, women's affairs and women's leagues, and the local health extension workers, with minimal participation of the women members. In Kilte-Awlaelo for example, the women members did not participate in the recruitment of their leaders', and the selected leaders were approved by the village council only if they were members of the council. Once selected, the 1 to 30 WDG leaders selected 1 to 5 WDG leaders in consultation with the women's associations, women's affairs, and women's leagues, and occasionally the village leaders. A majority of the respondents reported commitment, membership of village council (for 1 to 30 WDG leaders), and social acceptance as criteria to nominate and select WDG leaders. In all study areas, majority of the WDG leaders and health extension workers reported that there was no real community participation in the recruitment process. No value was given to their education level and marital status. Some also indicated that there were flaws in the nomination process.

"*What is amazing is the nomination process, which was held in the absence of the candidate. The candidate did not know what was going on, and no one informed her for two months after her election.*"

(In-depth interview, a 1 to 5 WDG leader)

*Training*. The recruited WDG leaders were supposed to receive orientation by their health extension workers on the health extension program packages for 60 hours. After that, they were expected to cascade this health orientation within their networks [10]. The WDG leaders indicated that the initial training was missing while others had got insufficient training.

*I have been working for more than six years. They have told us to start working but did not provide any training.*"

(Focus group discussion, a 1 to 5 WDG leader)

"*We haven't received any training in 2018 except one held at the woreda level last year.*"

(Focus group discussion, a 1 to 30 WDG leader)

The WDG leaders recommended training as a means of re-vitalizing the activities of the WDG leaders. The health extension workers and their supervisors reflected that although the initial training was provided, some newly replaced WDG leaders had not received this. Rather, they had been briefly oriented about women's affairs in general. The WDG leaders complained that trainings did not include sufficient refreshment, and there was no compensation for attending trainings. This hindered WDG leaders from fully attending training sessions.

*Incentives*. Participants repeatedly mentioned that at the start of the program, there were regular community forums that recognized the work of the WDG leaders. Such meetings were no longer organized at the time of the study. All health system informants pointed at the lacking criteria for providing incentives, and the difficulties in selecting the best performing WDG leaders. WDG leaders reported that the lack of incentives reduced their performance.

"*We are not incentivized for our work. I do not think that we should visit home-to-home daily for long hours without any incentive to cover the expenses for shoes.*"

(Focus group discussion, a 1 to 30 WDG leader)

The problematic selection of the WDG leaders and difficulties around training, recognition, and appreciation were frequently mentioned by WDG leaders as main reasons for low commitment. As a result, some also refused to volunteer.

"*On my side, I have almost stopped serving as a WDG leader. I have served for more than six years but they [Village leaders] have never sent me to training while others got a lot.*"

(In-depth interview, a 1 to 30 WDG leader)

The HEWs also stated that the 1 to 5 WDG leaders refused undertaking some activities because they did not want to do it for free. The WDG leaders expected to receive training, financial or in-kind incentives in return for their volunteer work.

**Support from community-based groups.** The WDG leaders mentioned many bodies or institutions that oversaw their activities. The national guideline states that the oversight of WDG program lies with women's affairs. The leaders, on the other hand, mentioned that they

were accountable to various actors, including the health sector, women's affairs, women's associations, and social affairs that perform activities with children and elderly, and the village administrator. Although they had irregular meetings, women's associations, women's affairs and women's leagues, and village administrators were mentioned to attend WDG leaders' meetings. They received poor support from the women's association, youth's association, farmer's association, male development groups, and other local institutions.

"*Due to lack of coordination, I don't think the male groups are supporting the WDG leaders' activities in our setting.*"

(Key informant interview, Woreda Health Extension Program expert)

The WDG leaders reported that elders, traditional healers, religious leaders, some traditional birth attendants, and married men supported their mobilization activities in the community. This support was based on the view that WDG leaders were trained and therefore trusted informants about maternal, neonatal and child health. In contrast, literate and young married women did not respect the WDG leaders and their activities.

"*Elders are our supporters, while literate and young women tend to undermine us. Once they told me it is because I do not have a husband that I am going home to home for no benefit. I became nervous at that time.*"

(IDI, a 1–30 WDG leader)

## Community health system performance outputs

**WDG leaders' competency.**   *Knowledge.* Some WDG leaders mentioned anemia, headache, swelling of face and hands, fever, bleeding, shivering, convulsions, and vaginal discharge as danger signs in pregnant women. They also talked about how to prepare food and feed children, and the need for vaccination and Vitamin A supplementation. A few WDG leaders also mentioned diarrhea, difficulty in breathing, fast breathing, coldness, fever, low birth weight, and vomiting as danger signs in newborn babies.

However, a majority of the WDG leaders frequently reported that they did not know newborn danger signs, or when to refer sick neonates. A 1 to 5 WDG leader said,

"*For example, a mother may tell me that a neonate is sick, but how could I know it. I have knowledge gaps here. What I would do is to refer it to the health facility.*"

*Service delivery.* Most WDG leaders reported that they identified people for maternal, neonatal and child health activities in their networks, including pregnant women, children, and also availability of hand-washing basins and latrines. The WDG leaders also informed pregnant women to attend the pregnant women conference, to visit antenatal care, and give birth at a health facility.

"*WDG leaders make home visits to tell pregnant women to attend follow-up care.*"

(Key informant interview, Health Extension Program expert)

Some WDG leaders also advised group members to undertake HIV-screening during pregnancy. WDG leaders also called ambulances when women in their groups were in labor.

"*I assure them that I will call ambulance for them if labor comes.*"

(Focus group discussion, a 1 to 30 WDG leader)

WDG leaders accompanied laboring women for delivery, although some ambulance drivers did not allow them to join. The WDG leaders also promoted and created awareness on the importance of delaying immediate bathing and initiating child immunization. They also advised women to initiate breastfeeding early, to exclusively breastfeed for the first six months, and to provide complementary feeding after six months. Some WDG leaders also reported that they identified and sent under-five children to the health facility for growth monitoring and food supplementation. They also participated in health campaigns like measles vaccination, vitamin A supplementation, deworming, and Zithromax distribution for prevention of trachoma. A few 1 to 30 WDG leaders also reported promoting bed net use, mobilizing resources to provide maternity care for poor women, and alleviating harmful traditional and gender-related practices.

Although all participants agreed that home visits and neonatal health care had been strengthened, only a few WDG leaders reported that they visited and assessed danger signs in mothers and the newborns.

"*Visiting homes has now been reduced compared to the time when the program started. If you ask women in the community, none of them would confirm to you that WDG leaders have visited them.*"

(Key informant interviews, HEW)

HEW supervisors reported that WDG leaders gave more attention to referring pregnant and laboring women as compared to sick neonates.

"*When we come to newborn care, I don't believe that we have given much attention to it. I do not think that the WDG leaders are working in this area.*"

(Key informant interview, Woreda maternal, neonatal and child health expert)

Woreda maternal, neonatal and child health experts also reported that WDG leaders rarely participated in promoting growth monitoring, Vitamin A supplementation, or deworming campaigns.

Overall, the HEW supervisors, woreda and regional key informants emphasized that the overall performance of the WDG leaders, particularly in the 1 to 5 networks, was very low.

"*If they [WDG leaders] work with what they know, the change would have been a lot. The problem is they are not committed to undertake the activities.*"

(Key informant interview, Health extension worker supervisor)

*Absenteeism*. The WDG leaders and health extension workers repeatedly mentioned that the WDG structure was getting weaker. In some villages, it had ceased to exist. In particular, the 1 to 5 networks were almost absent.

"*I would not say that there is work now. Of course, we were undertaking maternal related activities previously, but now it is almost deteriorating except for one or two of us.*"

(Focus group discussion, a 1 to 30 WDG leader)

Another leader also stated:

"*We were so strong in previous years. But currently, the members tend to restrict their work to private tasks at home.*"

The WDG leaders reported that their household chores, agriculture, social and development responsibilities prevented them from organizing and attending the WDG leaders' meetings. In addition, the WDG leaders were involved in non-health activities: agriculture, education, water, peace, security and justice, women's affairs, and women's associations. They were also involved in other social obligations and political activities, and collecting women's association membership fees in their communities. Moreover, the WDG leaders also reported pressure from their husbands not to work as volunteers.

**WDG leaders' wellbeing.**   *Motivation.* Some WDG leaders reported factors that motivated their work, which included a sense of duty.

"*I believe that I should serve my community. My parents undertook voluntary activities in their community and I would like to follow my parents' self-sacrifice.*"

(Focus group discussion, a 1 to 30 young WDG leader)

"*I am old and I am caring for an orphaned child, but still, I am doing my best as a 1 to 30 WDG leader in order not to break my promise to my big brother. He sacrificed his life for the betterment of the people of Tigray.*"

(In-depth interview, a 1 to 30 WDG leader)

Improving maternal and child health in the community was also said to inspire them to continue their voluntarism as a WDG leader. However, some WDG leaders complained that they were demotivated by the community's actions. For example, although they informed women to attend pregnant women's conferences, most women did not attend. Overall, they felt that the community refused to participate in the health extension program's packages.

The health extension workers had organized money saving groups within the WDG networks to motivate the WDG leaders to attend monthly meetings. However, these saving groups did not stay functional for long.

*Attrition.* The health extension workers' supervisors reported that due to poor linkages with the health extension workers, some of the WDG leaders were stressed, thereafter resigned from volunteering as leaders.

"*If you organized a meeting for WDG leaders and their members, you may get two out of five and six or eight out of the thirty, which makes it a tough problem to solve.*"

(Focus group discussion, health extension worker)

Many WDG leaders were either divorced or widowed. As a result, they feared of getting comments from the community, which forced them to resign.

"*I have presented my case to the village to replace me by someone younger who is able to read and write, an energetic woman; but the village leaders have asked me to stay in this volunteer service and I accepted for a few months.*"

(In-depth interview, a 1 to 30 WDG leader)

"*When I see myself, I do have 53 WDG leaders. Out of them, only 10 or 15 are continuing working as volunteers.*"

(Focus group discussion, health extension worker)

*Firing and replacement.* Firing of WDG leaders was carried out by women's associations and women's affairs. The reasons for firing a WDG leader included improper collection of women's association membership payments, assisting a pregnant woman to deliver at home, insufficient promotion of latrine construction, not using the recommended means of improving agriculture (e.g., using urea and selected seeds), and reported absence from meetings.

"*Let us say, if a woman leader ordered me to collect membership fees and I would not accept the order then they would conclude that I am not committed to work. I would immediately get a dismissal notice.*"

(Focus group discussion, a 1 to 30 WDG leader)

Some respondents reported that if the WDG leader was not considered as a model member of the community and not trusted by the community, she could be fired. The 1 to 30 WDG leaders also replaced 1 to 5 leaders, if they were dissatisfied with their performance. When WDG leaders were fired, resigned, moved, or passed away, there was usually a delay in finding replacements.

## Discussion

Women's Development Groups were established by the Ethiopian government to support the primary health care services for women, neonates, and children along with duties in other sectors [16]. We have shown that WDG leaders were committed to their health-related work. However, they were recruited in a sub-optimal process. Many were illiterate, had weak supervision and feedback from their health extension workers, lacked training and incentives, and had suboptimal knowledge on danger signs and care of neonates and children. These problems demotivated the volunteers and reduced their performance in promoting maternal, neonatal and child health. Health extension workers also faced difficulties in managing the large number of WDG leaders in their catchment area, while also noting the weakness of the leaders, particularly in the 1 to 5 networks.

The success of volunteer community health worker programs relies to a large degree on the people involved [15]. The main criteria for recruitment of the leaders were being part of a model family, trusted by members, and able to mobilize the community. We observed that the recruitment did not adhere to the WDG guidelines. Other research have underlined that social context and concerns raised by community members should be considered [15, 26]. These factors were not appraised in the recruitments. Neither their educational level, marital status nor their experience were considered in the selection of leaders [10]. These recruitment weaknesses may create community distrust in the WDG leaders' promotion of maternal, neonatal, and child health [15]. The recruitment criteria of the WDG leaders require revision to be flexible to local realities but sufficient to enhance trust in these voluntary workers [16].

Although WDG leaders recognized that they were supported by some community members and groups, they were not sufficiently supported and supervised by the health extension workers and health center staff. The core reasons for the limited support were the low commitment of staff at different levels of the primary health unit, and the large number of WDG leaders relative to the limited number of health extension workers [27]. There were also several government bodies overseeing the WDG leaders, although the direct oversite should have been with

the women's associations, women's affairs and women's leagues. Introducing a standardized supervision guideline for WDG leaders, reconfiguring the WDG network, and assigning an appropriate organization to oversee the network may improve the provision of support.

The absence of initial and regular refreshment training, particularly for the newly replaced WDG leaders, was reported in this study. In addition, resources were neither allocated to compensate WDG leaders for attending meetings, nor for refreshments during the meetings, which limited their attendance. The lack of training had implications for the WDG leaders' knowledge, which was reported to be particularly scanty on newborn health [2, 16]. Policy-makers and program experts involved in developing the training should consider regular trainings that address identified knowledge gaps and compensating WDG leaders for their training time.

At the time of the interview, there were no means of recognizing WDG leaders for their achievements. This fact, along with lack of other incentives, demotivated WDG leaders in their maternal, neonatal, and childcare activities. As a result, some refused volunteerism [14, 28]. Policy-makers and health managers should consider consistent, fair and locally-tailored motivation packages.

## Strengths and limitations

Using different data collection methods and interviewing participants across levels of the health system helped to observe convergence of ideas, highlighting factors that affect the WDG leaders' performance. The study participants had similar characteristics with other rural settings in Ethiopia, and therefore, some of the results may be transferable to similar settings in Ethiopia and beyond. The data analysis may have been limited by a possible loss of nuance during translation from Tigrigna to English. However, as far as possible, the translations were cross-checked by native Tigrigna-speaking research assistants who conducted the interviews and were fluent in English.

## Conclusion

Although the WDG program was created to provide a community link to the health extension services with maternal, neonatal and child healthcare to households, the current WDG leaders defined their work broader, i.e., as community development agents who also were responsible for activities outside the health sector. These ambiguities have led to poor management of the group. Furthermore, the WDG leaders' recruitment, supervision, and training did not follow national guidelines. Policy-makers should consider redesigning the program ensuring effective management as well as enhanced recruitment, training, supervision, and incentives amenable to local variations.

## Supporting information

**S1 File. Topic guide in English.**
(PDF)

**S2 File. Topic guide In Tigrigna.**
(PDF)

## Acknowledgments

We greatly acknowledge the Tigray regional health bureau and Mekelle University for their support. We also thank the Ethiopian Ministry of Health and the Dagu Project with team

members for their support. Special thanks to Dr Hagos Godefay and Zinabu Hadis for their technical advice. We like to express our appreciation to all participants and research assistants for their contributions and support in the field work. Finally, special thanks to the first author's relatives who were around and provided support in different ways during this study.

## Author Contributions

**Conceptualization:** Fisseha Ashebir, Araya Abrha Medhanyie, Afework Mulugeta, Lars Åke Persson, Della Berhanu.

**Data curation:** Fisseha Ashebir, Della Berhanu.

**Formal analysis:** Fisseha Ashebir.

**Funding acquisition:** Lars Åke Persson.

**Investigation:** Fisseha Ashebir, Araya Abrha Medhanyie, Afework Mulugeta, Lars Åke Persson.

**Methodology:** Fisseha Ashebir, Afework Mulugeta, Lars Åke Persson, Della Berhanu.

**Project administration:** Araya Abrha Medhanyie, Lars Åke Persson.

**Supervision:** Araya Abrha Medhanyie, Afework Mulugeta, Lars Åke Persson.

**Writing – original draft:** Fisseha Ashebir.

**Writing – review & editing:** Araya Abrha Medhanyie, Afework Mulugeta, Lars Åke Persson, Della Berhanu.

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
