## [Decision Letter · Decision Letter 0]

21 May 2021

PONE-D-21-08629

Exploring women’s development group leaders’ support to maternal, neonatal and child health care: a qualitative study in Tigray region, Ethiopia

PLOS ONE

Dear Dr. Berhanu,

Thank you for submitting your manuscript to PLOS ONE. After careful consideration, we feel that it has merit but does not fully meet PLOS ONE’s publication criteria as it currently stands. Therefore, we invite you to submit a revised version of the manuscript that addresses the points raised during the review process.

The academic editor served as a second reviewer. The Primary reviewer is an expert in the filed. The academic editor read agree that minor edits are warranted.

We look forward to receiving your revised manuscript.

Kind regards,

Joseph Telfair, DrPH, MSW, MPH

Academic Editor

PLOS ONE

Journal Requirements:

2. When reporting the results of qualitative research, we suggest consulting the COREQ guidelines: http://intqhc.oxfordjournals.org/content/19/6/349. In this case, please consider including more information on the number of interviewers, their training and characteristics; and on whether bias issues were considered.

3. Please provide additional details regarding participant consent. In the ethics statement in the Methods and online submission information, please ensure that you have specified how verbal consent was documented and witnessed.

Reviewers' comments:

Reviewer's Responses to Questions

**Comments to the Author**

1. Is the manuscript technically sound, and do the data support the conclusions?

Reviewer #1: Yes

2. Has the statistical analysis been performed appropriately and rigorously? 

Reviewer #1: N/A

3. Have the authors made all data underlying the findings in their manuscript fully available?

Reviewer #1: Yes

4. Is the manuscript presented in an intelligible fashion and written in standard English?

Reviewer #1: Yes

5. Review Comments to the Author

Reviewer #1: Thank you for the opportunity to review this manuscript on the WDG in Ethiopia.

I have a few minor suggested edits and a revision which I feel is important for overall understanding of the results.

Line 113 should be “by under-five children” not “of under-five children”

Lines 222-235 The recruitment paragraph in results is confusing and presents conflicting information? It starts by saying community groups are involved in selecting WDG. Then notes Klite-Awlaelo is an exception but not clear exception to what? The process which looks similar to what was just described? Then the paragraph closes saying communities were not involved in recruitment? Which conflicts with above? This is the one revision which is critical as there is also substantial discussion of this and it is a major conclusion and recommendation. So the results need to be more clear.

Line 269 delete “the” before “women’s”

Lines 283-284 The sentence result about literate young women would benefit from a quote to support this conclusion.

6. PLOS authors have the option to publish the peer review history of their article (what does this mean?). If published, this will include your full peer review and any attached files.

Reviewer #1: **Yes: **Debra Jackson

---

## [Author Response · Author response to Decision Letter 0]

10 Aug 2021

Editor’s comments

Journal Requirements:

Comment: Please review your reference list to ensure that it is complete and correct. If you have cited papers that have been retracted, please include the rationale for doing so in the manuscript text, or remove these references and replace them with relevant current references. Any changes to the reference list should be mentioned in the rebuttal letter that accompanies your revised manuscript. If you need to cite a retracted article, indicate the article’s retracted status in the References list and also include a citation and full reference for the retraction notice.

Response: Thank you for this comment. We have reviewed the reference list and ensured that it is complete and correct. We have not used references that have been retracted. The revised manuscript now follows the journal’s referencing requirements. 

Comment 1. Please ensure that your manuscript meets PLOS ONE's style requirements, including those for file naming. The PLOS ONE style templates can be found at https://journals.plos.org/plosone/s/file?id=wjVg/PLOSOne_formatting_sample_main_body.pdf and https://journals.plos.org/plosone/s/file?id=ba62/PLOSOne_formatting_sample_title_authors_affiliations.pdf

Response: Thank you for this comment. The manuscript has been revised according to the journal’s style requirements.

Comment 2. When reporting the results of qualitative research, we suggest consulting the COREQ guidelines: http://intqhc.oxfordjournals.org/content/19/6/349. In this case, please consider including more information on the number of interviewers, their training and characteristics; and on whether bias issues were considered.

Response: We thank you for your guidance. We have added more information on the interviewers and the type of training provided has been added to the manuscript See Methods section, on page 7, line 124. In order to cover diverse perspectives and experiences and reduce the risk of bias, we performed the study in different geographic areas and explored perceptions from different actors, i.e., active women’s development group leaders, health extension workers and managers (See Table 1). 

3. Please provide additional details regarding participant consent. In the ethics statement in the Methods and online submission information, please ensure that you have specified how verbal consent was documented and witnessed.

 Response: We thank you for this guidance. We have provided information on how verbal consent was documented in the manuscript (Methods section, page 9, line 164) and also on the online submission information.

Response: Thank you for this comment. We have added the following statement in the Data Availability section: “The qualitative data cannot be open access due to confidentiality reasons. De-identified data will be stored on s secure sever at the Ethiopian Publica Health Institute and can be made available upon request by contacting Martha Zeweldemariam martha.zeweldemariam@lshtm.ac.uk.” Page 23 line 475.

Response: We thank you for your guidance. The supporting information captions have been included after the references on page 25.

Reviewers' comments:

Reviewer's Responses to Questions

Comments to the Author

1. Is the manuscript technically sound, and do the data support the conclusions?

Reviewer #1: Yes

2. Has the statistical analysis been performed appropriately and rigorously?

Reviewer #1: N/A

3. Have the authors made all data underlying the findings in their manuscript fully available?

Reviewer #1: Yes

4. Is the manuscript presented in an intelligible fashion and written in standard English?

Reviewer #1: Yes

5. Review Comments to the Author

Reviewer #1: Thank you for the opportunity to review this manuscript on the WDG in Ethiopia.

I have a few minor suggested edits and a revision which I feel is important for overall understanding of the results.

Response to the Reviewer:

Reviewer #1 

Line 113 should be “by under-five children” not “of under-five children”

Response: We thank you for this comment; the change has been made accordingly (page 6, line 106). 

Lines 222-235 the recruitment paragraph in results is confusing and presents conflicting information? It starts by saying community groups are involved in selecting WDG. Then notes Kilte-Awlaelo is an exception but not clear exception to what? The process which looks similar to what was just described? Then the paragraph closes saying communities were not involved in recruitment? Which conflicts with above? This is the one revision which is critical as there is also substantial discussion of this and it is a major conclusion and recommendation. So, the results need to be more clear. 

Response: We thank for these comments. We have adjusted the paragraph so it is clear that women members were not involved in the selection and used the Kilte -Awlaelo as an example for their lack of participation (page 11 line 216).

Line 269 delete “the” before “women’s”

Response: We thank you for this comment; the change has been made accordingly (Page 14, line 267).

Lines 283-284 The sentence result about literate young women would benefit from a quote to support this conclusion.

Response: We thankful for your valid support, quote has been added to the manuscript (Page 14, line 283). 

6. PLOS authors have the option to publish the peer review history of their article (what does this mean?). If published, this will include your full peer review and any attached files.

Do you want your identity to be public for this peer review? For information about this choice, including consent withdrawal, please see our Privacy Policy.

Reviewer #1: Yes: Debra Jackson

---

## [Decision Letter · Decision Letter 1]

7 Sep 2021

Exploring women’s development group leaders’ support to maternal, neonatal and child health care: a qualitative study in Tigray region, Ethiopia

PONE-D-21-08629R1

Dear Dr. Berhanu,

We’re pleased to inform you that your manuscript has been judged scientifically suitable for publication and will be formally accepted for publication once it meets all outstanding technical requirements.

Kind regards,

Joseph Telfair, DrPH, MSW, MPH

Academic Editor

PLOS ONE

Additional Editor Comments (optional):

The academic Editor served as the second reviewer for this manuscript and agree it should be accepted. 

Reviewers' comments:

Reviewer's Responses to Questions

**Comments to the Author**

1. If the authors have adequately addressed your comments raised in a previous round of review and you feel that this manuscript is now acceptable for publication, you may indicate that here to bypass the “Comments to the Author” section, enter your conflict of interest statement in the “Confidential to Editor” section, and submit your "Accept" recommendation.

Reviewer #1: All comments have been addressed

2. Is the manuscript technically sound, and do the data support the conclusions?

Reviewer #1: Yes

3. Has the statistical analysis been performed appropriately and rigorously? 

Reviewer #1: Yes

4. Have the authors made all data underlying the findings in their manuscript fully available?

Reviewer #1: Yes

5. Is the manuscript presented in an intelligible fashion and written in standard English?

Reviewer #1: Yes

6. Review Comments to the Author

Reviewer #1: All comments addressed.

I have no competing interests

interesting findings on maternal health support groups.

7. PLOS authors have the option to publish the peer review history of their article (what does this mean?). If published, this will include your full peer review and any attached files.

Reviewer #1: **Yes: **Debra Jackson

---

## [Editor Report · Acceptance letter]

16 Sep 2021

PONE-D-21-08629R1 

Exploring women’s development group leaders’ support to maternal, neonatal and child health care: a qualitative study in Tigray region, Ethiopia 

Dear Dr. Berhanu:

I'm pleased to inform you that your manuscript has been deemed suitable for publication in PLOS ONE. Congratulations! Your manuscript is now with our production department. 

Kind regards, 

on behalf of

Dr. Joseph Telfair 

Academic Editor

PLOS ONE